# CHIPNET: BUDGET-AWARE PRUNING WITH HEAVISIDE CONTINUOUS APPROXIMATIONS

**Rishabh Tiwari**[†‡*]**, Udbhav Bamba**[†‡*]**, Arnav Chavan**[†‡*] **and Deepak K. Gupta**[†**]
[†]Transmute AI Research, The Netherlands
[‡]Indian Institute of Technology, ISM Dhanbad, India
[*]Informatics Institute, University of Amsterdam, The Netherlands

## ABSTRACT

Structured pruning methods are among the effective strategies for extracting small resource-efficient convolutional neural networks from their dense counterparts with minimal loss in accuracy. However, most existing methods still suffer from one or more limitations, that include 1) the need for training the dense model from scratch with pruning-related parameters embedded in the architecture, 2) requiring model-specific hyperparameter settings, 3) inability to include budget-related constraint in the training process, and 4) instability under scenarios of extreme pruning. In this paper, we present *ChipNet*, a deterministic pruning strategy that employs continuous Heaviside function and a novel *crispness loss* to identify a highly sparse network out of an existing dense network. Our choice of continuous Heaviside function is inspired by the field of design optimization, where the material distribution task is posed as a continuous optimization problem, but only discrete values (0 or 1) are practically feasible and expected as final outcomes. Our approach's flexible design facilitates its use with different choices of budget constraints while maintaining stability for very low target budgets. Experimental results show that ChipNet outperforms state-of-the-art structured pruning methods by remarkable margins of up to 16.1% in terms of accuracy. Further, we show that the masks obtained with ChipNet are transferable across datasets. For certain cases, it was observed that masks transferred from a model trained on feature-rich teacher dataset provide better performance on the student dataset than those obtained by directly pruning on the student data itself. [1]

## 1 INTRODUCTION

Convolution Neural Networks (CNNs) have resulted in several breakthroughs across various disciplines of deep learning, especially for their effectiveness in extracting complex features. However, these models demand significantly high computational power, making it hard to use them on low-memory hardware platforms that require high-inference speed. Moreover, most of the existing deep networks are heavily over-parameterized resulting in high memory footprint (Denil et al., 2013; Frankle & Carbin, 2018). Several strategies have been proposed to tackle this issue, that include network pruning (Liu et al., 2018), neural architecture search using methods such as reinforcement learning (Jaafra et al., 2019) and vector quantization (Gong et al., 2014), among others.

Among the methods outlined above, network pruning has proved to be very effective in designing small resource-efficient architectures that perform at par with their dense counterparts. Network pruning refers to removal of unnecessary weights or filters from a given architecture without compromising its accuracy. It can broadly be classified into two categories: *unstructured pruning* and *structured pruning*. Unstructured pruning involves removal of neurons or the corresponding connection weights from the network to make it sparse. While this strategy reduces the number of parameters in the model, computational requirements are still the same (Li et al., 2017). Structured pruning methods on the other hand remove the entire channels from the network. This strategy pre-

---

[*]All authors contributed equally.
[1]Code is publicly available at https://github.com/transmuteAI/ChipNet

serves the regular structure, thereby taking advantage of the high degree of parallelism provided by modern hardware (Liu et al., 2017; Gordon et al., 2018).

Several structured pruning approaches have been proposed in the recent literature. A general consensus is that variational approaches using sparsity prior loss and learnable dropout parameters outperform the deterministic methods (Lemaire et al., 2019). Some of these methods learn sparsity as a part of pretraining, and have proved to perform better than the three stage pretrain-prune-finetune methods. However, since such approaches need to train the model from scratch with pruning-related variables embedded into the network, they cannot benefit from off-the-shelf pretrained weights (Liu et al., 2017; Alvarez & Salzmann, 2017). Others require choosing hyperparameters based on the choice of the network, and cannot be easily adapted for new models (Gordon et al., 2018). Further, with most of these methods, controlled pruning cannot be performed, and a resource-usage constraint can only be satisfied through trial-and-error approach. Recently, Lemaire et al. (2019) presented a budget-aware pruning method that includes the budget constraint as a part of the training process. A major drawback of this approach and other recent methods is that they are unstable for very low resource budgets, and require additional tricks to work. Overall, a robust budget-aware pruning approach that can be coupled with different budget constraints as well as maintains stability for very low target budgets, is still missing in the existing literature.

In this paper, we present *ChipNet*, a deterministic strategy for structured pruning that employs continuous Heaviside function and crispness loss to identify a highly sparse network out of an existing pretrained dense network. The abbreviation 'ChipNet' stands for Continuous Heaviside Pruning of Networks. Our pruning strategy draws inspiration from the field of design optimization, where the material distribution task is posed as a continuous optimization problem, but only discrete values (0 or 1) are practically feasible. Thus, only such values are produced as final outcomes through continuous Heaviside projections. We use a similar strategy to obtain the masks in our sparsity learning approach. The flexible design of ChipNet facilitates its use with different choices of budget constraints, such as restriction on the maximum number of parameters, FLOPs, channels or the volume of activations in the network. Through experiments, we show that ChipNet consistently outperforms state-of-the-art pruning methods for different choices of budget constraints.

ChipNet is stable for even very low resource budgets, and we demonstrate this through experiments where network is pruned to as low as 1% of its parameters. We show that for such extreme cases, ChipNet outperforms the respective baselines by remarkable margins, with a difference in accuracy of slightly beyond 16% observed for one of the experiments. The masks learnt by ChipNet are transferable across datasets. We show that for certain cases, masks transferred from a model trained on feature-rich teacher dataset provide better performance on the student dataset than those obtained by directly pruning on the student data itself.

## 2 RELATED WORK

As has been stated in the hypothesis by Frankle & Carbin (2018), most neural networks are overparameterized with a large portion (as much as 90%) of the weights being of little significance to the output of the model. Clearly, there exists enormous scope to reduce the size of these networks. Several works have explored the efficiency of network pruning strategies for reducing storage requirements of these networks and accelerating inference speed (LeCun et al., 1990; Dong et al., 2017). Some early works by Han et al. (2015a;c); Zhu & Gupta (2017) involve removal of individual neurons from a network to make it sparse. This reduces the storage requirements of these networks, however, no improvement in inference speed is observed. Recently, several works have focused on structured network pruning, as it involves pruning the entire channel/filters or even layers to maintain the regular structure (Luo et al., 2017; Li et al., 2017; Alvarez & Salzmann, 2016).

The focus of this paper is on structured network pruning, thus, we briefly discuss here the recent works related to this approach. The recent work by Li et al. (2017) identifies less important channels based on L1-norm. Luo et al. (2017); He et al. (2017) perform channel selection based on their influence on the activation values of the next layer. Liu et al. (2017) perform channel-level pruning by imposing LASSO regularization on the scaling terms in the batchnorm layers, and prune the model based on a global threshold. He et al. (2018b) automatically learn the compression ratio of each layer with reinforcement learning. Louizos et al. (2017); Alvarez & Salzmann (2017; 2016) train and prune the network in a single stage strategy.

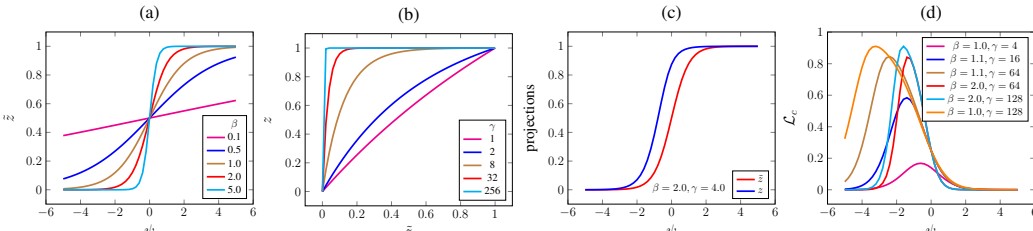

Figure 1: Representation of the different functions used in ChipNet for various choices of $\beta$ and $\gamma$. The plots show (a) logistic curves, (b) continuous Heaviside functions, (c) the outputs of logistic and Heaviside functions shown together for $\beta = 2.0$ and $\gamma = 4.0$, and (d) the crispness loss function.

The above mentioned approaches cannot optimize networks for a pre-defined budget constraint. Adding budget constraint to the pruning process can provide a direct control on the size of the pruned network. For example, Morphnet imposes this budget by iteratively shrinking and expanding a network through a sparsifying regularizer and uniform layer wise width multiplier, respectively, and is adaptable to specific resource constraints (Gordon et al., 2018). However, it requires a model-specific hyperparameter grid search for choosing the regularization factor. Another approach is BAR (Lemaire et al., 2019) that uses a budget-constrained pruning approach based on variational method. A limitation of this approach is that for low resource budgets, it needs to explicitly ensure that at least one channel is active in the downsample layer to avoid *fatal pruning*. The approach proposed in this paper does not require any such tweaking, and is stable for even very low resource budgets.

## 3 PROPOSED APPROACH

### 3.1 LEARNING SPARSITY MASKS

*Sparsity learning* forms the core of our approach. It refers to learning a set of sparsity masks for a dense convolutional neural network (*parent*). When designing the smaller pruned network (*child*), these masks identify the parts of the parent that are to be included in the child network. We first describe here the general idea of learning these masks in the context of our method.

The proposed approach falls in the category of structured pruning where masks are designed for channels and not individual neurons. Let $f : \mathbb{R}^d \to \mathbb{R}^k$ denote a convolutional neural network with weights $\mathbf{W} \in \mathbb{R}^m$ and a set of hidden channels $\mathbf{H} \in \mathbb{R}^p$. We define $\mathbf{z} \in \mathbb{R}^p$ as a set of sparsity masks, where $z_i \in \mathbf{z}$ refers to the mask associated with the feature map $\mathbf{h}_i \in \mathbf{H}$. To apply the mask, $z_i$ is multiplied with all the entries of $\mathbf{h}_i$. The optimization problem can further be stated as

$$\min_{\mathbf{W}, \mathbf{z}} \mathcal{L}(f(\mathbf{z} \odot \mathbf{H}(\mathbf{W}); \mathbf{x}), \mathbf{y}) \ \text{ s.t. } \ \mathcal{V}(\mathbf{z}) = \mathcal{V}_0, \tag{1}$$

where $\odot$ denotes elementwise multiplication and $\{\mathbf{x}, \mathbf{y}\} \in \mathcal{D}$ are data samples used to train the network $f$. The desired sparsity of the network is defined in terms of the equality constraint, where $\mathcal{V}(\cdot)$ denotes the budget function and $\mathcal{V}_0$ is the maximum permissible budget. Our proposed formulation of pruning is independent from the choice of the budget function. We later show this through experiments with volume budget as in Lemaire et al. (2019), channel budget similar to Liu et al. (2017), and budget defined in terms of parameters and FLOPs as well.

Originally, $z_i \in \mathbf{z}$ would be defined such that $z_i \in \{0, 1\}$, and a discrete optimization problem is to be solved. For the sake of using gradient-based methods, we convert it to a continuous optimization problem, such that $z_i \in [0, 1]$. Such reformulation would lead to intermediate values of $z$ occurring in the final optimized solution. Any intermediate value of $z$, for example $z = 0.4$, would imply that a fraction of the respective channel is to be used, and clearly such a solution is practically infeasible. We propose to overcome this challenge through the use of simple nonlinear projections and a novel loss term, and these are discussed in detail in the next section.

### 3.2 Continuous Heaviside Approximation and Logistic Curves

At the backbone of our pruning strategy lies three important functions: the commonly used *logistic curves*, *continuous Heaviside function* and *crispness loss* term. Figure 1 presents a graphical representation of these functions. We further provide below a brief motivation for the choice of these functions as well as their significance in our pruning approach.

*Logistic curves.* A commonly used function for adding nonlinearity to a neural network (LeCun et al., 1998), logistic curve projects an input from the real space to a range of 0-1 (Figure 1a), and can be mathematically stated as

$$\tilde{z} = \frac{1}{1 + e^{-\beta(\psi - \psi_0)}}, \tag{2}$$

where $\psi$ denotes the optimization parameter corresponding to the mask $z$, $\psi_0$ is the midpoint of the curve, and $\tilde{z}$ denotes the resultant intermediate projection. The additional parameter $\beta$ is used to control the growth rate of the curve, and forms an important ingredient of our approach. While low values of $\beta$ can produce an approximately linear curve between -1 and 1, higher values turn it into a step function. During the initial stages of training, we propose to keep $\beta$ very low, and increase it to higher values at later stages of the optimization process. With increased values of $\beta$, the values further from 0.5 are made more favorable for $\tilde{z}$.

In our experience, the logistic curve alone cannot be used to obtain approximately discrete (0-1) solutions for $\mathbf{z}$ in a continuous optimization scheme. The nonlinearity introduced by this function cannot sufficiently penalize the intermediate values between 0 and 1, and optimization algorithm can easily identify values of $\psi$ for which the projected values are far from both. An example experiment demonstrating this issue is presented in Appendix C.2. To circumvent this issue, we add another nonlinear projection using a continuous approximation of the Heaviside function.

*Continuous Heaviside function.* A continuous approximation to the Heaviside step function, referred as continuous Heaviside function in this paper, is a commonly used projection strategy to solve continuous versions of binary optimization (0-1) problems in the domain of design optimization (Guest et al., 2004; 2011). The generalized form of this function can be stated as:

$$z = 1 - e^{-\gamma \tilde{z}} + \tilde{z}e^{-\gamma}, \tag{3}$$

where, the parameter $\gamma$ dictates the curvature of the regularization. Figure 1b shows the continuous Heaviside function for several values of $\gamma$. We see that $z$ is linear for $\gamma = 0$ and approaches the Heaviside step function for very large values of $\gamma$.

The advantages of our projection function are twofold. First, during projection, the values close to 0 and 1 are not affected irrespective of the choice of $\gamma$. This implies that the masks identified with most confidence in the early stage of training are not directly impacted by the continuation applied on the value of $\gamma$, thus helping in the convergence of the training process. Here, 'continuation' refers to slowly adapting the value of $\gamma$ during the course of training. Second, even the values of $\tilde{z}$ which are slightly greater than 0 are also nonlinearly projected to close to 1, and this effect is more prominent for larger values of $\gamma$. The projection adds higher penalty on values between 0 and 1, and makes them extremely unfavorable when higher values of $\gamma$ are chosen.

While the continuous Heaviside function helps to obtain approximately discrete masks, there is still no explicit constraint or penalty function that can regulate this. To overcome this problem, we tie the outputs of logistic and continuous Heaviside functions to define a novel loss term, referred as crispness loss.

*Crispness Loss.* This novel loss term explicitly penalizes the model performance for intermediate values of $\mathbf{z}$, and drives the convergence towards crisp (0-1) masks. It is defined as the squared $L_2$ norm of the difference between $\tilde{\mathbf{z}}$ and $\mathbf{z}$, stated as $\mathcal{L}_{\mathbf{c}} = \|\tilde{\mathbf{z}} - \mathbf{z}\|_2^2$, and from Figure 1c, we see that $\mathcal{L}_c$ achieves its minima when either $\tilde{z} = z = 0$ or $\tilde{z} = z = 1$. Further, the trend of this loss function with respect to $\psi$ for different values of $\beta$ and $\gamma$ is shown in Figure 1d. It can be seen that for lower values of $\beta$ and $\gamma$, the loss value is low, and the crispness function plays little to no role in driving the pruning process. When the value of $\gamma$ slowly increases, the peak of the graph shifts upwards as well as towards the left, thereby increasing the penalty associated with values of $\psi$. This drives the values of $\psi$ to move farther from the origin. The left shift in the graph adds higher penalty on the negative values, forcing them to become even more negative, thus forcing the respective $z$ to move closer to 0.

The additional loss function associated with the model generally favors values towards 1. For example, cross-entropy loss used for classification would prefer to set all values in $\mathbf{z}$ to 1 to be able to maximize the classification accuracy. With increasing values of $\gamma$ forcing the masks towards 0, a balance between the two is identified during the training process. The term $\beta$ acts as a regularizer that to some extent counteracts the too abrupt impact of $\gamma$ and regulates the convergence of the training process.

### 3.3 IMPOSING BUDGET CONSTRAINT

The simplicity of our pruning strategy decouples it from the choice of budget constraint. In this paper, we show the working with four different choices of budget constraints: *channel, activation volume, parameters* and *FLOPs*. These choices are inspired from some of the recent state-of-the-art methods from the existing literature (Liu et al., 2017; Lemaire et al., 2019).

For budget calculation, values of the masks $\mathbf{z}$ should be close to 0 or 1. However, during the initial iterations of training, masks would contain intermediate values as well. This makes it difficult to accurately calculate the budget for the constraint specified in Eq. 1. Thus, rather than computing it directly over the masks $\mathbf{z}$, these are computed on $\bar{\mathbf{z}}$, where $\bar{z}_i \in \bar{\mathbf{z}}$ is obtained by applying a logistic projection on $\mathbf{z}$ with $\psi_0 = 0.5$ (Eq. 2). Further discussion related to it is provided in Appendix C.3.

The budget constraint is imposed using a loss term $\mathcal{L}_b$, referred as *budget loss*. We define the budget loss as $\mathcal{L}_b = (\mathcal{V}(\mathbf{z}) - \mathcal{V}_0)^2$, where $\mathcal{V}(\cdot)$ can be one of the 4 budget functions described below.

*Channel budget.* It refers to the maximum number of hidden channels $\mathbf{h}$ that can be used across all convolutional layers of the network. Mathematically, it can be stated as $\mathcal{V}^{(c)} = (\sum_{i=1}^{p} \bar{z}_i)/p$, where $p$ denotes the number of hidden channels in the network. Constraint on the channel budget limits the number of channels, and thus the number of weights in the network.

*Volume budget.* This budget controls the size of the activations, thereby imposing an upper limit on the memory requirement for the inference step. We define volume budget $\mathcal{V}^{(v)} = (\sum_{j=1}^{\mathcal{N}(h)} \sum_{i=1}^{p_j} A_j \bar{z}_i)/(\sum_{j=1}^{\mathcal{N}(h)} A_j \cdot p_j)$, where $\mathcal{N}(h)$ denotes the number of convolutional layers, and $A_j$ and $p_j$ denote area of the feature maps and their count, respectively, in the $j^{\text{th}}$ layer.

*Parameter budget.* This budget directly controls the total number of parameters in the network, and can thus be used to impose an upper limit on the size of the model. For details, see Appendix A.1.

*FLOPs budget.* This budget can be directly used to control the computational requirement of the model. Mathematical formulae to calculate it is stated in Appendix A.1.

### 3.4 SOFT AND HARD PRUNING

The pruning stage in our approach comprises two steps: *soft pruning* and *hard pruning*. After a deep dense network has been pretrained, masks are added to the network and soft pruning is performed. The steps involved in soft pruning are stated in Algorithm 1. During this stage, the network is optimized with the joint loss $\mathcal{L} = \mathcal{L}_{ce} + \alpha_1 \mathcal{L}_c + \alpha_2 \mathcal{L}_b$, where $\alpha_1$ and $\alpha_2$ are the weights for the crispness and budget loss terms, respectively.

After every epoch of soft pruning, the performance of the network is evaluated in a hard pruned manner. For this purpose, masks $\mathbf{z}$ are used, and a cutoff is chosen using binary search such that the budget constraint is exactly satisfied. Values above this cutoff are converted to 1 and the ones lower turned to 0. Finally, the model with best performance on the validation set is chosen for fine tuning.

---

**Algorithm 1:** ChipNet Pruning Approach

---

**Input** : pretrained network weights $\mathbf{W}$;
budget constraint function $\mathcal{V}(\cdot)$;
budget value $\mathcal{V}_0$; training data $\mathcal{D}$;
pruning iterations $N$

**Output:** learnt sparsity masks $\mathbf{z}$

$\psi_i \in \Psi \leftarrow$ random initialization

**for** $k = 1 \ldots N$ **do**

    $(\mathbf{x}, \mathbf{y}) \leftarrow \texttt{sample}(\mathcal{D})$

    $\tilde{\mathbf{z}} \leftarrow \text{LOGISTIC}(\psi)$

    $\mathbf{z} \leftarrow \text{CONTINUOUSHEAVISIDE}(\tilde{\mathbf{z}})$

    $\hat{\mathbf{y}} \leftarrow \texttt{Forward}(\mathbf{x}, \mathbf{W}, \mathbf{z})$ $\mathcal{V} \leftarrow \mathcal{V}(\mathbf{z})$

    $\mathcal{L} \leftarrow \text{CHIPNETLOSS}(\mathcal{V}, \mathcal{V}_0, \tilde{\mathbf{z}}, \mathbf{z}, \hat{\mathbf{y}}, \mathbf{y})$

    $(\nabla \mathbf{W}, \nabla \psi) \leftarrow \texttt{Backward}(\mathcal{L})$

    $(\mathbf{W}, \psi) \leftarrow$
    $\texttt{OptimizeStep}(\nabla \mathbf{W}, \nabla \psi)$

**end**

---

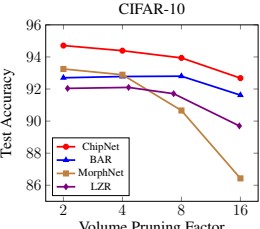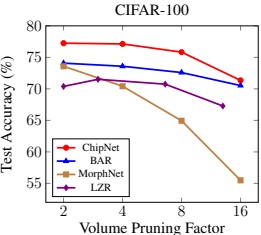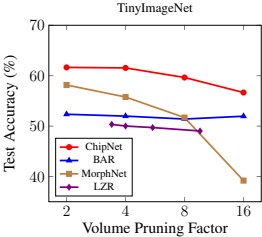

Figure 2: Performance comparison of ChipNet with different structured pruning baselines for various choices of volume constraint. Here, volume pruning factor refers to the factor by which the volume budget is being reduced.

## 4 EXPERIMENTS

### 4.1 EXPERIMENTAL SETUP

We test the efficacy of our pruning strategy on several network architectures for four different choices of budget constraint functions. The architectures chosen in this study include WideResNet-26-12 (WRN-26-12) (Zagoruyko & Komodakis, 2016), PreResNet-164 (He et al., 2016b), ResNet-50 and ResNet-101 (He et al., 2016a). For datasets, we have chosen CIFAR-10/100 (Krizhevsky, 2009) and Tiny ImageNet (Wu et al.). For the combined loss $\mathcal{L}$ in Eq. 1, weights $\alpha_1$ and $\alpha_2$ are set to 10 and 30, respectively, across all experiments. Implementation details related to the pretraining, pruning and finetuning steps, as well details of the hardware are described in Appendix B.

### 4.2 RESULTS

**Performance of pruned sparse networks.** We present here results obtained using ChipNet for WRN-26-12 and PreResNet-164 pruned with volume and channel constraints, respectively. For the two other constraints, parameters and FLOPs budget, we perform a comparative study later in this paper.

*Volume budget.* Figure 2 shows the comparison of performance values for WRN-26-12 for CIFAR-10, CIFAR-100 and TinyImageNet datasets, when pruned using ChipNet. We compare our results with BAR (Lemaire et al., 2019), MorphNet (Gordon et al., 2018) and LZR (Louizos et al., 2017) for volume pruning factors of 2, 4, 8 and 16. Details related to the three baselines are presented in Appendix D.1. ChipNet consistently outperforms all the baselines across all datasets and for all choices of the budget. For the case of extreme pruning of 16 folds on CIFAR-100, the performance of BAR is close to ours, while the other two baselines significantly underperform.

*Channel budget* We study here the pruning efficacy of ChipNet coupled with channel constraint on PreResNet-164 architecture for CIFAR-10 and CIFAR-100 datasets. Results are compared with the network slimming approach (Liu et al., 2017), implementation details related to which can be found in Appendix D.1. As constraints, we use channel budgets of 60%, 40%, 20% and 10%.

Table 1 presents the results for different choices of channel budgets. We also report the number of parameters in the pruned network as well as the associated FLOPs. It is seen that ChipNet outperforms the baseline method for all the experimental settings. For CIFAR-10 in particular, we see that for even very low channel budget of 10%, accuracy of the pruned network drops by only 3.1%. For 10% channel budget, our method outperforms the network slimming strategy on CIFAR-10 and CIFAR-100 by remarkable margins of 8.5% and 16.1%, respectively.

Note that lower channel usage does not necessarily imply lower number of parameters or reduced FLOPS in the pruned network, and we analyze this for the various cases of pruning considered in Table 1. We see that ChipNet performs selection of channels in a more optimized way, such that better accuracy is achieved with fewer parameters. In terms of FLOPS, both methods perform at par. Although, the FLOPS for ChipNet are slightly higher for the channel budget of 10%, this overhead is insignificant compared to the gain in accuracy and reduction of parameters. Overall, we infer that ChipNet couples well with the channel constraint, and is stable for even extreme pruning cases of as low as 10% channel budget.

Table 1: Performance scores for pruning PreResNet-164 architecture on CIFAR-10 and CIFAR-100 datasets for Network Slimming and ChipNet (ours). The number of parameters and FLOPs for the unpruned networks are 1.72 million and $5.03 \times 10^8$, respectively. Here budget refers to the percentage of total channels remaining. Abbreviations 'Acc.' and 'Params.' refer to accuracy and number of parameters, all scores are reported in %, and parameters and FLOPs are reported relative to those in the unpruned network.

| Method | Budget(%) | CIFAR-10 | | | CIFAR-100 | | |
|---|---|---|---|---|---|---|---|
| | | Acc. ↑ | Params. ↓ | FLOPs ↓ | Acc. ↑ | Params. ↓ | FLOPs ↓ |
| Unpruned | - | 94.9 | 100.0 | 100.0 | 77.1 | 100.0 | 100.0 |
| Net-Slim | 60 | 95.3 | 85.1 | 79.0 | 77.5 | 85.9 | **75.1** |
| ChipNet | | 95.3 | **79.3** | **77.9** | **77.8** | **85.0** | 75.2 |
| Net-Slim | 40 | 94.9 | 65.4 | 58.9 | 76.6 | 71.9 | 55.4 |
| ChipNet | | **95.0** | **51.7** | **54.7** | **77.3** | **65.8** | **53.1** |
| Net-Slim | 20 | 93.0 | 33.3 | 29.9 | 70.1 | 44.7 | 25.0 |
| ChipNet | | **94.2** | **24.0** | **28.4** | **72.3** | **31.8** | **23.9** |
| Net-Slim | 10 | 87.1 | 19.0 | **15.3** | 51.2 | 19.2 | **11.1** |
| ChipNet | | **91.8** | **13.8** | 16.4 | **67.3** | **14.6** | 12.6 |

Table 2: Performance scores for pruning ResNet-50 architecture on CIFAR-100 and CIFAR-10 for BAR and ChipNet (ours) with volume budget (V) and channel budget (C). The number of parameters and FLOPS for the unpruned networks are 23.7 million and $2.45 \times 10^9$, respectively. Here budget refers to the percentage of total channels/volume remaining. Abbreviations 'Acc.' and 'Param.' refer to accuracy and number of parameters, all scores are reported in %, and parameters and FLOPs are reported relative to those in the unpruned network.

| Method | Budget (%) | CIFAR-10 | | | CIFAR-100 | | |
|---|---|---|---|---|---|---|---|
| | | Acc. ↑ | Param. ↓ | FLOPs ↓ | Acc. ↑ | Param. ↓ | FLOPs ↓ |
| Unpruned | - | 93.3 | 100 | 100 | 73.0 | 100 | 100 |
| ChipNet (C) | | 92.8 | 4.5 | 17.7 | 71.1 | 7.3 | 10.9 |
| ChipNet (V) | 12.5 | 91.0 | 2.8 | 5.1 | 65.5 | 22.5 | 9.0 |
| BAR (V) | | 88.4 | 1.8 | 3.8 | 63.8 | 5.2 | 4.2 |
| ChipNet (C) | | 92.1 | 1.6 | 8.8 | 67.0 | 1.8 | 4.8 |
| ChipNet (V) | 6.25 | 83.6 | 1.3 | 2.0 | 54.7 | 14.5 | 5.1 |
| BAR (V) | | 84.0 | 0.9 | 1.3 | 42.9 | 3.7 | 2.0 |

*Effect of the choice of budget.* Here, we analyze the impact of one budget type over another to understand whether the choice of budget really matters when pruning a network. As a first experiment, we study side-by-side the results for channel and volume constraints when used to prune ResNet-50 on CIFAR-10 and CIFAR-100 datasets. Results of this experiment are shown in Table 2. Note that we do not intend to identify a winner among the two, since both are meant to optimize different aspects of the network. For baseline comparison, the network is also pruned using the BAR method. The volume budget variant of ChipNet outperforms BAR by a significant margin. Moreover, we see that for the same amount of volume constraint, the number of parameters used by BAR are lower than our method for most cases. A reason for significant drop in performance of BAR could be that the optimization algorithm does not fully exploit the choice of channels to be dropped, thereby choosing a sub-optimal set and losing too many parameters from the network.

Between the results of volume and channel constraints for ChipNet, at the first look, it seems that channel constraint is better throughout. However, as stated above, a direct comparison between the two is unfair. For example, volume constraints are meant to reduce the number of activations, and in turn would also reduce the FLOPs. This is evident from the results as FLOPS reported for volume constraint are always lower than the respective channel constraint. For a better understanding of the effects of these budgets, we perform another experiment for pairwise analysis of these constraints.

Figure 3: Test accuracy versus the remaining budget for networks pruned using ChipNet with different budget constraints.

Figure 3 shows the pairwise plots of the budgets used to prune WRN-26-12 on CIFAR-100. From the first two plots, we see that the scores reported are higher for any volume budget when the network is optimized with volume constraint, and similarly higher for a certain channel budget when the network is optimized for it. Similar observations can also be made between the number of parameters and FLOPs. In a nutshell, we observe that the pruned network performs best with respect to the constraint for which the masks are trained. Thus, the choice of constraint type should not be arbitrary but based on the practical applications, such as reducing FLOPs, among others.

*Stability and robustness.* Our pruning strategy is also very stable, and this has already been demonstrated earlier for channel and volume pruning at low resource budgets. Compared to the baselines, networks obtained with ChipNet are found to perform significantly better even without the need for any additional tweaking such as explicitly opening certain channels to ensure network connectivity (Liu et al., 2017; Lemaire et al., 2019). Another example demonstrating the stability is volume pruning (6.25%) of ResNet-50 on CIFAR-100, where ChipNet performs 11.8% better than BAR.

To account for robustness, we have extensively performed hyperparameter grid search on a channel budget of 6.25% for WRN-26-12 to identify the suitable values for $\alpha_1$ and $\alpha_2$. It has been observed that values in the neighborhood of this point do not affect the performance. Details related to this grid search are further provided in Appendix C.5. Further, the same hyperparameter setting has been used for all the experiments. The consistent results across all datasets shows that ChipNet is robust.

**Transfer learning of masks.** Inspired by knowledge distillation (Hinton et al., 2015), where refined information obtained from a deeper teacher network is transferred to a shallow student network, we study here the transfer of sparsity masks across datasets. For teacher and student, we use Tiny ImageNet and CIFAR-100 datasets, respectively, and ResNet-101 is pruned for different choices of channels budgets. Table 8 reports the performance scores for the pruned network on CIFAR-100 when the masks are learnt on CIFAR-100 as well as when they are learnt on Tiny ImageNet and transferred. Interestingly, for moderate channel budgets of 40% and 60%, we see that the models using masks transferred from Tiny ImageNet perform better than those obtained directly on CIFAR-100. This gain in performance from mask transfer could be attributed to the feature-richness of the chosen teacher

Table 3: Accuracy values (%) on CIFAR-100 dataset for ResNet-101 pruned with different choices of channel budget (%) on CIFAR-100 (Base) and with masks from Tiny ImageNet (Transfer).

| Budget | Base Acc | Transfer Acc |
|--------|----------|--------------|
| 20 | **71.3** | 68.3 |
| 40 | 71.6 | **72.0** |
| 60 | 71.8 | **72.1** |
| 100 | 73.6 | - |

dataset. We also see that for the very low budget case of 20%, masks from the student dataset outperform that from the teacher. For such low budgets, the expressive power of the model is too low to fully exploit the knowledge from the transferred masks.

## 5 CONCLUSION

We have presented ChipNet, a deterministic strategy for structured pruning of CNNs based on continuous Heaviside function and crispness loss. Our approach provides the flexibility of using it with different budget constraints. Through several experiments, it has been demonstrated that ChipNet outperforms the other methods on representative benchmark datasets. We have also shown that

ChipNet can generate well-performing pruned architectures for very low resource budgets as well. To conclude, with the strongly effective pruning capability that ChipNet exhibits, it can be used by the machine learning community to design efficient neural networks for a variety of applications.

## 6 Limitations and Future Work

In this paper, we have explored the stability and robustness of ChipNet from various perspectives. Through experiments, we have shown that ChipNet consistently performs well across several CNN architectures and datasets. We analyzed it with respect to different choices of budget constraints, performed stability tests for even extreme scenarios of as low as 1% parameters, analyzed how the masks get distributed across the network, and even studied the transferability of masks. For all these experiments, ChipNet has proved to work well. However, before ChipNet can be considered a full-proof solution for pruning, additional experiments might be needed. For example, the applicability of ChipNet is not yet explored on large datasets such as ImageNet, and we would like to explore it in our future work.

Further, it would be of interest to explore how the pruned architectures obtained using ChipNet perform for tasks beyond classification, such as segmentation and object tracking. Improving inference speed is an important aspect of object tracking, and it has not been explored from the point of view of network pruning. We would like to see if the recent object tracking algorithms that use backbones such as ResNet-50 and ResNet-101 can be made faster through our pruning method.

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

APPENDICES

# A  EXTENSION: PROPOSED APPROACH

## A.1  BUDGET CONSTRAINTS

Additional details related to the 4 budget constraints discussed in this paper follow below.

*Channel budget.*  It refers to the maximum number of hidden channels $\mathbf{h}$ that can be used across all convolutional layers of the network. Mathematically, it can be stated as

$$\mathcal{V}^{(c)} = \frac{\sum_{i=1}^{p} \bar{z}_i}{p},\tag{4}$$

where $p$ denotes the number of hidden channels in the network.

*Volume budget.*  This budget controls the size of the activations, thereby imposing an upper limit on the memory requirement for the inference step. We define volume budget as

$$\mathcal{V}^{(v)} = \frac{\sum_{j=1}^{\mathcal{N}(h)} \sum_{i=1}^{p_j} A_j \bar{z}_i}{\sum_{j=1}^{\mathcal{N}(h)} A_j \cdot p_j},\tag{5}$$

where $\mathcal{N}(h)$ denotes the number of convolutional layers in the network, and $A_j$ and $p_j$ denote area of the feature maps and their count, respectively, in the $j^{\text{th}}$ layer.

*Parameter budget.*  This budget term directly controls the total number of parameters in the network, and can thus be used to impose an upper limit on the size of the model parameters. It is defined as

$$\mathcal{V}^{(p)} = \frac{\sum_{j=1}^{\mathcal{N}(h)} (K_j \cdot \sum_{i=1}^{p_j} \bar{z}_i^j \cdot \sum_{i=1}^{p_{j-1}} \bar{z}_i^{j-1} + 2 \cdot \sum_{i=1}^{p_j} \bar{z}_i^j)}{\sum_{j=1}^{\mathcal{N}(h)} (K_j \cdot p_j \cdot p_{j-1} + 2 \cdot p_j)},\tag{6}$$

where $K_j$ denotes area of the kernel. The two terms in the numerator account for the number of parameters in the convolutional layer and the batchnorm layer.

*FLOPs budget.*  This budget can be directly used to control the computational requirement of the model. Assuming that a sliding window is used to achieve convolution and the nonlinear computational overhead is ignored, the FLOPs budget of the convolution neural network can be defined as in Molchanov et al. (2016):

$$\mathcal{V}^{(f)} = \frac{\sum_{j=1}^{\mathcal{N}(h)} (K_j \cdot \sum_{i=1}^{p_{j-1}} \bar{z}_i^{j-1} + 1) \cdot \sum_{i=1}^{p_j} \bar{z}_i^j \cdot A_j}{\sum_{j=1}^{\mathcal{N}(h)} (K_j \cdot p_{j-1} + 1) \cdot p_j \cdot A_j}.\tag{7}$$

# B  TRAINING PROCEDURE

Details regarding the pretraining, pruning and finetuning steps are discussed below:

## B.1  PRE-TRAINING

**WRN-26-12, MobileNetV2, ResNet-50, ResNet-101, ResNet-110** were trained with batch size of 128 at initial learning rate of $5 \times 10^{-2}$ using SGD optimizer with momentum 0.9 and weight decay $10^{-3}$. We use step learning rate strategy to decay learning rate by 0.5 after every 30 epochs. For CIFAR-10 and CIFAR-100, models were trained for 300 epochs whereas for Tiny ImageNet the number of epochs were reduced to half to have maintain same number of iterations.

**Preresnet-164** was trained with batch size of 64 at initial learning rate of $\times 10^{-1}$ using SGD optimizer with momentum 0.9 and weight decay $10^{-4}$. We use Multi Step Learning rate strategy to decay learning rate by 0.1 after $80^{th}$ and $120^{th}$ epoch. The model was trained for 160 epochs for all datasets. This strategy is adopted from Liu et al. (2017)

### B.2 Pruning

A common pruning strategy was applied for all models irrespective of budget type or dataset. AdamW (Loshchilov & Hutter, 2019) with constant learning rate of $10^{-3}$ and weight decay of $10^{-3}$ was used as optimizer. Weight decay for $\psi$ was kept 0. Weight for budget loss and crispness loss is kept constant to 30 and 10 respectively. $\beta$ increases by $2 \times 10^{-2}$ after every epoch starting from 1 and $\gamma$ doubles after every 2 epochs starting from 2.

### B.3 Fine-Tuning

The finetuning of pruned model is done exactly similar to the pre-training step.

### B.4 Hardware

All experiments were run on a Google Cloud Platform instance with a NVIDIA V100 GPU (16GB), 16 GB RAM and 4 core processor.

## C Additional Experiments

### C.1 Pruning with volume and channel budget

This section shows results of ChipNet along with different baselines pruned with with channel and volume budget. Table 4 is an extension to Table 2 presented in section 4.2. Table 5 shows numerical values corresponding to figure 2 discussed in section 4.2.

Table 4: Performance scores for pruning ResNet-50 architecture on CIFAR-100/CIFAR-10 for BAR and ChipNet (ours) with volume budget (V) and channel budget (C). The number of parameters and FLOPS for the unpruned networks are 23.7 million and $2.45 \times 10^9$, respectively. Abbreviations 'Acc.' and 'Param.' refer to accuracy and number of parameters, all scores are reported in %, and parameters and FLOPs are reported relative to those in the unpruned network.

| Method | Budget (%) | CIFAR-10 | | | CIFAR-100 | | |
|---|---|---|---|---|---|---|---|
| | | Acc. ↑ | Param. ↓ | FLOPs ↓ | Acc. ↑ | Param. ↓ | FLOPs ↓ |
| Unpruned | - | 93.3 | 100 | 100 | 73.0 | 100 | 100 |
| ChipNet (C) | | 93.1 | 36.5 | 58.8 | **72.7** | 44.1 | 40.9 |
| ChipNet (V) | 50 | **93.4** | 18.6 | 29.0 | 72.1 | 58.0 | 38.4 |
| BAR (V) | | 91.4 | 9.5 | 21.3 | 71.5 | 22.5 | 24.9 |
| ChipNet (C) | | **93.0** | 12.3 | 30.0 | **72.6** | 18.7 | 20.7 |
| ChipNet (V) | 25 | 92.9 | 4.9 | 12.3 | 69.9 | 32.1 | 17.2 |
| BAR (V) | | 91.5 | 2.3 | 7.4 | 71.8 | 5.4 | 7.3 |
| ChipNet (C) | | **92.8** | 4.5 | 17.7 | **71.1** | 7.3 | 10.9 |
| ChipNet (V) | 12.5 | 91.0 | 2.8 | 5.1 | 65.5 | 22.5 | 9.0 |
| BAR (V) | | 88.4 | 1.8 | 3.8 | 63.8 | 5.2 | 4.2 |
| ChipNet (C) | | **92.1** | 1.6 | 8.8 | **67.0** | 1.8 | 4.8 |
| ChipNet (V) | 6.25 | 83.6 | 1.3 | 2.0 | 54.7 | 14.5 | 5.1 |
| BAR (V) | | 84.0 | 0.9 | 1.3 | 42.9 | 3.7 | 2.0 |

### C.2 Pruning with only Logistic Curves

As discussed in section 3.2, continuous Heavyside approximation helps to penalize intermediate values of $\mathbf{z}$ to attain values closer to 0-1. Only with logistic curves the distribution of soft masks gets concentrated at one point as shown in Figure 4b. Although, the budget constraint will be satisfied, this kind of distribution hinders effective channel selection as the relative importance of $\mathbf{z}$ cannot be determined concretely. Contrary to this, using heaviside function with crispness loss models $\mathbf{z}$ in

Table 5: Performance scores for pruning WideResNet architecture on CIFAR-10, CIFAR-100 and Tiny ImageNet datasets for BAR (Lemaire et al., 2019), MorphNet (Gordon et al., 2018), ID (Denton et al., 2014), WM (Han et al., 2015b), Random Pruning and ChipNet (ours). All results are reported in % accuracy

| Method | Budget (%) | CIFAR-10 ↑ | Tiny ImageNet ↑ | CIFAR-100 ↑ |
|---|---|---|---|---|
| BAR | 50 | 92.7 | 52.4 | 74.1 |
| | 25 | 92.8 | 52 | 73.6 |
| | 12.5 | 92.8 | 51.4 | 72.6 |
| | 6.25 | 91.6 | 52.0 | 70.5 |
| MorphNet | 50 | 93.3 | 58.2 | 73.6 |
| | 25 | 92.9 | 55.8 | 70.4 |
| | 12.5 | 90.7 | 51.7 | 69.9 |
| | 6.25 | 86.4 | 39.2 | 55.5 |
| ID | 50 | 91.09 | 49.96 | 69.29 |
| | 25 | 91.44 | 49.55 | 69.75 |
| | 12.5 | 90.37 | 45.77 | 66.03 |
| | 6.25 | 86.92 | 39.72 | 59.13 |
| WM | 50 | 91.11 | 49.01 | 68.98 |
| | 25 | 91.20 | 49.67 | 69.10 |
| | 12.5 | 89.68 | 47.72 | 65.42 |
| | 6.25 | 86.33 | 40.19 | 58.99 |
| Random | 50 | 89.63 | 48.25 | 67.51 |
| | 25 | 88.02 | 46.08 | 63.64 |
| | 12.5 | 84.62 | 39.41 | 59.22 |
| | 6.25 | 81.36 | 29.53 | 48.88 |
| ChipNet | 50 | 94.7 | 61.6 | 77.3 |
| | 25 | 94.4 | 61.5 | 77.1 |
| | 12.5 | 93.9 | 59.6 | 75.8 |
| | 6.25 | 92.7 | 56.7 | 71.4 |

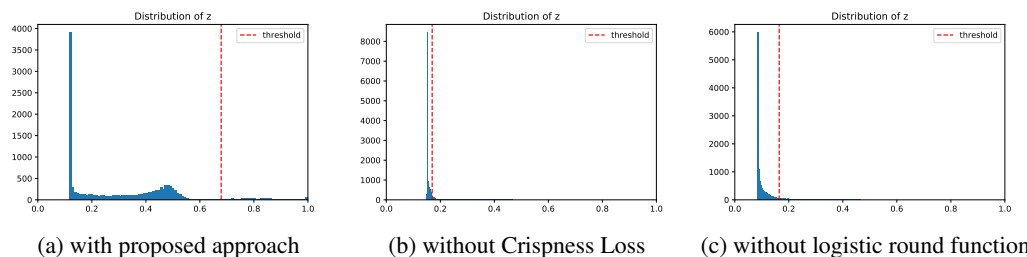

| (a) with proposed approach | (b) without Crispness Loss | (c) without logistic round function |
|---|---|---|

Figure 4: Distribution of zetas obtained on pruning WRN-26-12 with CIFAR-100 for 16x channel pruning factor.

terms of their relative importance as shown in Figure 4a and hence results in more effective pruning of less important channels.

### C.3 ROLE OF LOGISTIC-ROUNDING IN BUDGET CALCULATION

As discussed in section 3.3 budget calculation is done on $\bar{z}$ rather than computing it directly over the masks $\mathbf{z}$ where $\bar{z}_i \in \bar{\mathbf{z}}$ is obtained by applying a logistic projection on $\mathbf{z}$ with $\psi_0 = 0.5$ (Eq. 2). Importance of this projection can be seen through figure 4c. The distribution of soft masks obtained with the proposed approach (Figure 4a) is clearly much more distinct than the one calculated without logistic round projection (Figure 4c). Thus a better threshold can be selected to choose the active sparsity mask that satisfies the budget constraint.

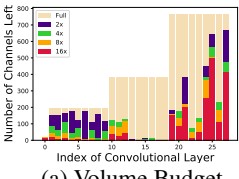 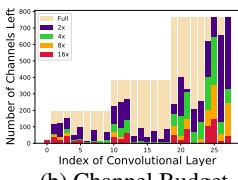 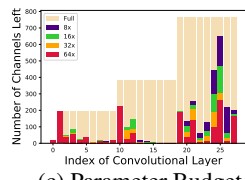 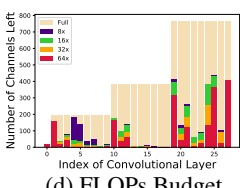

(a) Volume Budget     (b) Channel Budget     (c) Parameter Budget     (d) FLOPs Budget

Figure 5: Visualization of the number of channels remaining per convolutional layer in the architectures obtained from ChipNet with different choices of budget constraint and various pruning factors.

## C.4 Effect of the choice of budget

We further visualize how ChipNet performs pruning across the various convolutional layers of a network for different choices of budget. Figure 5 shows the number of active channels per convolutional layer for several pruning factors for the 4 budget types. These results have been obtained on WRN-26-12. We see that the pruned networks with low resource budgets are aligned with those with the higher budgets in terms of distribution of active channels across layers. This could mean that the networks pruned for low resource budgets should be achievable hierarchically from those pruned on larger budgets. Further, we also see that there are layers with almost no channels left. The performance of our model is still not affected since these dead layers correspond to the skip connections in the network. ChipNet identifies such extra connections, and eliminates them if the performance of the model is not affected significantly.

## C.5 Hyperparameter grid search

An extensive grid search is done to search hyperparameters, $\alpha_1$, $\alpha_2$, $b_{inc}$ and $g_{inc}$ for pruning WRN-26-12 on CIFAR-100 at 6.25% volume budget. Here $\alpha_1$ and $\alpha_2$ are the weightage values given to crispness loss and budget loss repectively in the joint loss as shown in section 3.4. $\beta_{inc}$ and $\gamma_{inc}$ refers the the number of epochs after which value of beta increases by 0.1 and value of gamma doubles, effect of these hyperparameters is discussed in section 3.2. We found a cloud of values for which the pruning accuracy is comparable. This cloud can be seen in table 6. We choose the best values from these for all our other experiments as we concluded that model pruning is less senstive to these hyperparameters.

Table 6: Grid search on WRN-C100 for 16x volume pruning factor. Here Acc refers to the validation accuracy of hard pruned model during pruning.

| $\alpha_1$ | $\alpha_2$ | $\beta_{inc}$ | $\gamma_{inc}$ | **Acc(%)** |
|---|---|---|---|---|
| 10 | 30 | 5 | 2 | 5.5 |
| 10 | 45 | 5 | 1 | 5.4 |
| 15 | 30 | 1 | 1 | 5.3 |
| 15 | 30 | 5 | 1 | 5.2 |
| 10 | 30 | 5 | 1 | 4.8 |
| 10 | 60 | 5 | 1 | 4.7 |
| 15 | 20 | 1 | 1 | 4.7 |
| 5 | 60 | 2 | 2 | 4.6 |
| 15 | 60 | 5 | 2 | 4.5 |
| 1 | 45 | 1 | 1 | 4.5 |
| 15 | 30 | 5 | 2 | 4.3 |
| 5 | 45 | 2 | 2 | 4.2 |
| 15 | 60 | 2 | 1 | 3.9 |
| 5 | 45 | 5 | 1 | 3.9 |
| 10 | 20 | 1 | 1 | 3.8 |

## C.6 Senstivity Analysis

In this section we show the senstivity analysis for WRN-26-12 on Cifar-100 at 16x Volume budget constraint. We ran 5 experiments of pruning where value of all four hyperparameters $\alpha_1$, $\alpha_2$, $\beta_{inc}$, $\gamma_{inc}$ were sampled from the uniform distribution with -+10% perturbations from the tuned values.

Table 7: Senstivity analysis on WRN-C100 for 16x volume pruning factor. Here Accuracy refers to the test accuracy of hard pruned model after finetuning.

| $\alpha_1$ | $\alpha_2$ | $\beta_{inc}$ | $\gamma_{inc}$ | **Accuracy** |
|---|---|---|---|---|
| 10.19 | 28.23 | 5.12 | 1.98 | 0.7143 |
| 10.34 | 29.17 | 4.67 | 2.05 | 0.7232 |
| 9.28 | 27.34 | 4.8 | 1.91 | 0.72 |
| 9.24 | 32.25 | 5.13 | 1.89 | 0.7132 |
| 9.24 | 30.47 | 5.48 | 2.07 | 0.7168 |
| | | | Mean | 0.7175 |
| | | | Std dev. | 0.00412 |

## C.7 Transferability of Mask

Here we show the complete results of Table 3 to depict the transferability of mask proposed in section 4.2

Table 8: Accuracy values (%) on CIFAR-100 dataset for ResNet-101 pruned with different choices of channel budget (%) on CIFAR-100 (Base) and with masks from Tiny ImageNet (Host).

| Budget(%) | Tiny ImageNet (Host Acc) | C100 (Base Acc) | C100 (Transfer Acc) |
|---|---|---|---|
| 20 | 51.6 | **71.3** | 68.3 |
| 40 | 55.2 | 71.6 | **72.0** |
| 60 | 56.0 | 71.8 | **72.1** |
| 100 | 63.3 | 73.6 | - |

## D Implementation details

### D.1 Baseline methods

**BAR, LZR, MorphNet, WM, ID on WRN-26-12**: All results are taken from Lemaire et al. (2019). We reproduced a few results to cross-check and ensure that there are no big deviations. We found that our reproduced results were very close to the one reported in the paper.

**BAR on Resnet-50**: Results are produced from the code given by (Lemaire et al., 2019). Pruning strategy is adopted from Lemaire et al. (2019) and the number of iterations are adjusted to match ours for fair comparison.

**Network Slimming on PreResNet-164**: We have reproduced the results by using the same pre-training, pruning and finetuning strategy is used by Liu et al. (2017) and the same pretraining and finetuning strategy is used for our results in order to do fair comparison of pruning algorithms.

## E Pseudo code

Here we present the explanations and pseudo-codes for the various functions used in 1.

### E.1 LOGISTIC FUNCTION

---
**Algorithm 2:** LOGISTIC

---
**Input** : Optimization parameter corresponding to every mask $\psi$; Growth rate control
        parameter $\beta$

**Output:** Resultant intermediate projection $\tilde{\mathbf{z}}$

$\tilde{\mathbf{z}} \leftarrow \frac{1}{1+e^{-\beta\psi}}$

---

### E.2 CONTINUOUS HEAVISIDE FUNCTION

---
**Algorithm 3:** CONTINUOUS HEAVISIDE

---
**Input** : Intermediate projection $\tilde{\mathbf{z}}$; Curvature regularization parameter $\gamma$

**Output:** Resultant final projection $\mathbf{z}$

$\mathbf{z} \leftarrow 1 - e^{-\gamma\tilde{z}} + \tilde{z}e^{-\gamma}$

---

### E.3 CHIPNET LOSS FUNCTION

---
**Algorithm 4:** CHIPNET LOSS

---
**Input** : Target budget $\mathcal{V}_0$; Current model budget $\mathcal{V}$; Intermediate projection $\tilde{\mathbf{z}}$; Final projection
        $\mathbf{z}$; Predicted output $\hat{\mathbf{y}}$; Ground truth $\mathbf{y}$; Crispness loss weight $\alpha_1$; Budget loss weight
        $\alpha_2$

**Output:** Loss $\mathcal{L}$

$\mathcal{L}_{ce} \leftarrow -\sum \mathbf{y}\log(\hat{\mathbf{y}})$

$\mathcal{L}_c \leftarrow \|\tilde{\mathbf{z}} - \mathbf{z}\|_2^2$

$\mathcal{L}_b \leftarrow (\mathcal{V} - \mathcal{V}_0)^2$

$\mathcal{L} \leftarrow \mathcal{L}_{ce} + \alpha_1 \mathcal{L}_c + \alpha_2 \mathcal{L}_b$

---

### E.4 FORWARD FUNCTION

Forward function takes three inputs - network weights ($\mathbf{W}$), input data batch ($\mathbf{x}$) and sparsity masks ($\mathbf{z}$). The forward function is the forward pass of regular CNN; with one change that the respective sparsity masks are multiplied to the activation obtained after every batch normalization layer.

### E.5 BACKWARD FUNCTION

The backward function is the back propagation pass of a regular CNN to obtain the gradients of the loss with respect to the model parameters ($\mathbf{W}$ and $\psi$ ).

## F ADDITIONAL RESULTS

Table 9: Performance scores for pruning MobileNetV2 architecture on Cifar-10 for ChipNet with channel budget.

| Budget(%) | Acc. ↑ |
|---|---|
| Unpruned | 93.55 |
| 80 | 92.58 |
| 60 | 92.44 |
| 40 | 91.98 |
| 20 | 90.65 |

Table 10: Performance scores for pruning ResNet-50 architecture on Tiny-ImageNet for ChipNet with volume budget (V) and channel budget (C).

| Method | Budget(%) | Acc. ↑ |
|--------|-----------|--------|
| Unpruned | - | 61.38 |
| ChipNet (C) | 50 | 56.65 |
|  | 25 | 54.72 |
|  | 12.5 | 52.73 |
|  | 6.25 | 47.49 |
| ChipNet (V) | 50 | 54.23 |
|  | 25 | 53 |
|  | 12.5 | 50.03 |
|  | 6.25 | 45.51 |

Table 11: Accuracy values (%) on CIFAR-10 dataset for ResNet-110 pruned using volume budget.

| Model | Base acc. ↑ | Prune acc. ↑ | FLOPs reduction ↑ |
|-------|-------------|--------------|-------------------|
| Pruning-A (Li et al., 2016) | 93.53% | 93.51% | 1.19x |
| Pruning-B (Li et al., 2016) | 93.53% | 93.30% | 1.62x |
| SFP (He et al., 2018a) | 93.68% | 93.86% | 1.69x |
| C-SGD-5/8 (Ding et al., 2019) | 94.38% | 94.41% | 2.56x |
| CNN-FCF-A (Li et al., 2019) | 93.58% | 93.67% | 1.76x |
| CNN-FCF-B (Li et al., 2019) | 93.58% | 92.96% | 3.42x |
| Group-HS 7e-5 (Yang et al., 2019) | 93.62% | 94.06% | 2.30x |
| Group-HS 1e-4 (Yang et al., 2019) | 93.62% | 93.80% | 3.09x |
| Group-HS 1.5e-4 (Yang et al., 2019) | 93.62% | 93.54% | 4.38x |
| Group-HS 2e-4 (Yang et al., 2019) | 93.62% | 92.97% | 5.84x |
| Chip Net (Volume-2x) | 93.98% | 93.78% | 2.66x |
| Chip Net (Volume-4x) | 93.98% | 92.38% | 7.54x |
| Chip Net (Volume-8x) | 93.98% | 91.36% | 12.53x |

