# OpenReview forum: "ChipNet: Budget-Aware Pruning with Heaviside Continuous Approximations"
_ICLR.cc/2021/Conference — ICLR 2021 Poster_

### Official Review · AnonReviewer3 · 2020-10-26

**Rating:** 6
**Confidence:** 4

**Review:**

##########################################################################


Summary:

The paper provides a budget-aware regularizer method to train the network to be pruned. Existing methods based-on the regularizer method suffer from satisfying the user-specified constraints and resort to trial-and-error approach. The authors leverage logistic function and continuous heaviside function for continuous relaxation of the discrete variable to select the channels to be pruned.  There are four types of budget constraints (channel, volume, parameter, FLOPs).
The experiment results on several different budget constraints (even in extreme condition) are quite promising. Also, these methods can be applied in transfer learning (transferrable network structure).

##########################################################################


Reasons for score:

Overall, I vote for marginal acceptance. I like the budget-aware constraints concept and its tightness which is shown in figure 3.
However, I have several concerns about the papers (see cons) and authors need to consider these concerns in the rebuttal period.


##########################################################################

1. The budget-aware constraint is explicitly introduced and easy to implement.

2. Ablation study in figure 5 provides insights into the network channel pruning.

##########################################################################


Cons:

(1) Experiment results about the other several pruning experiments are omitted. Therefore, I concern that the other readers are confused about the performance of the proposed method over the other pruning methods. Particularly, I wish to check the performance with [1].  Also, Experiments on the ImageNet (large dataset experiments) should be added for future research.

(2) Details on the forward steps are omitted. (See 'Forward(x,W,z)' in Algorithm 1) Readers should have prior knowledge to guess the details on the forwarding steps.

(3) Justification of the usage of two functions (logistic and heaviside function) is not enough. Two separate functions for continuous approximations are quite confusing. For example, '(Logistic(\psi)+1)/2' can be used for continuous relaxation or 'Gumbel trick' can be used.

(4) This might be the most critical one. Since the budget constraint is given based on the l2 loss with the target budget, '\alpha2' should be tuned like the other regularizer based method. From a critical perspective,  there is no difference with the other regularized-based method in the sense that the trial-and-error on the tuning '\alpha2' is stilled required and very large '\alpha2' could degrade the performance of the network.


=======================================================================================================

After rebuttal :

Sorry for the delay. The authors address my concerns and reflect them on the revised version. I'm keeping my rating.


[1] DeepHoyer: Learning Sparser Neural Network with Differentiable Scale-Invariant Sparsity Measures, ICLR 2020.

---

> ### Author Response · Authors · 2020-11-16
> **Response to Reviewer #3**
>
> Thank you for your constructive comments. We appreciate that you find our experimental results promising.
>
> **Ans1**
> We would like to mention that our approach is a budget-aware structured pruning method, and only very few previous works have addressed this aspect of pruning. Thus, we mostly focused on methods that actually handle budget constraints or can be adapted for the same in a structured pruning setting. As asked by the reviewer, we have now also performed a comparison with DeepHoyer [1] for ResNet110 and CIFAR10. We found the performance of our method comparable to DeepHoyer, however, we reiterate that one benefit of our method is the ability to handle budget constraints. Results related to this are added in Section F of the updated draft.
>
> Additionally, we have now also added performance scores for [2], [3], and Random Baseline on CIFAR-10, CIFAR-100 and TInyImageNet using WRN-26-12. Details related to these can be found in Tables 5 of Appendix C.1 in the updated draft. Also, we agree that the performance of ChipNet should be explored on popular models such as ResNet50 and large datasets such as ImageNet. Given the limited time of rebuttal, adding the results of ImageNet would not be feasible, and we will pursue it as a part of our future research work. For the sake of completeness, we have added results related to ResNet50 on TinyImageNet in Appendix F of the new draft. To the best of our knowledge, there exists no previous work that has shown pruning on such a combination. Thus, a quantitative analysis is hard to make. Nevertheless, we have observed that ChipNet is stable for this pruning problem and the resultant model from pruning of ResNet50 on TinyImageNet shows similar behaviour as other experiments of the paper.
>
> ################################################################################
>
> **Ans2**
> Thank you for pointing this out. We have now updated the draft to include pseudo-codes or additional descriptions for each of the functions mentioned in Algorithm 1. These can be found in Appendix E in the updated draft.
>
> ################################################################################
>
> **Ans3**
> We have expressed in detail the motivation for choosing each of the two functions. However, we agree with the reviewer that the justification for the combined use of both of these might not be clear in the paper, so we will clarify it here. We agree that continuous relaxation can also be described using a single function, such as those specified by the reviewer. However, we would like to highlight two limitations of deterministic methods that comprise a single continuous function. First, it is hard to achieve crisp solutions with single projection functions. In our experience, the optimizer can always find solutions for the mask that are neither close to 1 nor 0, irrespective of the extent of nonlinearity added to the projections. This is also described in the ablation study presented in Appendix C.2. Using the logistic and Heaviside functions together offers a simple yet effective strategy to define the ‘crispness loss’, a crucial function for designing discrete solutions using the continuous optimization approach.
>
> ################################################################################
>
> **Ans4**
> We would like to mention that unlike the other regularizer based methods, the performance of ChipNet is not very sensitive to the choice of $\alpha_2$. We demonstrate this quantitatively in the stability study presented in Appendix C.5. From the interpretation perspective, $\alpha_2$ does not control much the performance of our pruning strategy since it is not directly applied on the pruning masks, but only used to weigh the budget loss in the overall loss function. This is justified further from the fact that unlike other regularizer-based methods such as Network Slimming [4], we use a single value of $\alpha_2$ throughout all the experiments reported in the paper. We hope to have sufficiently answered this question. In case you would like to understand more, we would be happy to elaborate further.
>
> ################################################################################
>
> [1] DeepHoyer: Learning Sparser Neural Network with Differentiable Scale-Invariant Sparsity Measures, ICLR 2020.
>
> [2] Learning both weights and connections for efficient neural network, NIPS, 2015
>
> [3] Exploiting Linear Structure Within Convolutional Networks for Efficient Evaluation, NIPS, 2014
>
> [4] Learning Efficient Convolutional Networks through Network Slimming, ICCV, 2017

---

### Official Review · AnonReviewer1 · 2020-10-27
**Review on Paper1752**

**Rating:** 7
**Confidence:** 3

**Review:**

Summary

This paper presents a new method for structure pruning called ChipNet. The ChipNet employs continuous Heaviside function with commonly used logistic curve and crispness loss to estimate sparsity masks. A combination of above three components is helpful to obtain approximately discrete solutions for a continuous optimization scheme.  As a result, it is possible to get a highly sparse network out of an existing pre-trained dense network. Experimentally, ChipNet outperforms other previous structured pruning methods by a large margin.

Overall,  I don't think this paper has any critical drawbacks, and there are only a few comments. Therefore, I recommend accepting this paper.

Strength

The motivation for this paper is clear and quite interesting. Also, the performance of the proposed pruning method is much better than previous baseline methods, and experiments were performed in various ways to show the effectiveness of it.



Comments and Weakness

I have not much to criticize this paper, but I have some questions and comments as follows.

Is it possible to get a similar result by applying the proposed method to the originally lightweight network such as Mobile Net?

Are there comparisons with NAS-based methods?

Is the proposed method applicable to networks for other tasks such as detection and segmentation?


There is a submitted paper (to this ICLR) that includes continuous relaxation of discrete network structure optimization for network growing (not pruning).

“Growing Efficient Deep Networks by Structured Continuous Sparsification”

The direction of the paper is totally different, but there seems to be a similar part in terms of continuous relaxation. It would be good to mention the difference with this paper w.r.t. continuous relaxation.

Also, I would like that more related works are mentioned in recent techniques using continuous relaxation(approximation).

---

> ### Author Response · Authors · 2020-11-19
> **Response to Reviewer #1**
>
> Thank you for your constructive feedback. We are very happy that you found our method interesting and effective.
>
> **Q1.** Is it possible to get a similar result by applying the proposed method to the originally lightweight network such as Mobile Net?
>
> **Ans** We experimented with Mobilenetv2 on CIFAR 10 dataset with channel budget and the results are now available in Appendix F in the updated draft. The pruning pattern and convergence behavior seem similar to other experiments. Since there is no official baseline in our knowledge, we compare our results with an unofficial implementation of Network Slimming [1] and L1-norm pruning [2] available at [3].
>
> | Method | Budget(%) | Fine Tuning Acc(%) |
> |---------|-----|----------|
> | [1] |  | 92.01 |
> | [2] | 60 | 91.81 |
> | Our | | 92.44 |
> | | | |
> | [1] |  | 91.50 |
> | [2] | 40 | 84.40 |
> | Our | | 91.98 |
>
> [1] Learning Efficient Convolutional Networks through Network Slimming, ICCV, 2014
>
> [2] Pruning Filters for Efficient ConvNets, ICLR, 2017
>
> [3] https://github.com/wlguan/MobileNet-v2-pruning
>
> ---
>
> **Q2.** Are there comparisons with NAS-based methods?
>
> **Ans** Thank you for this remark. We have not made a comparison with NAS implementations. However, since this might be a fair comparison for better understanding the efficacy of ChipNet, we will consider it in our future work.
>
> ---
> **Q3.** Is the proposed method applicable to networks for other tasks such as detection and segmentation?
>
> **Ans** This is a very interesting point. It would be of interest to explore how pruning methods perform on tasks such as detection, segmentation and object tracking. More importantly, we believe object tracking is an interesting field to test the potential of pruning methods. There exist object tracking methods that use detection and segmentation methods as a part of their pipeline. We are already working to extend the applicability of this method for object tracking, and a segmentation task.
>
> Currently,one of the SOTA single object visual object tracking algorithms, namely, Siam-RCNN [4] uses ResNet-101 backbone which is first pretrained on ImageNet and then trained on videos. However, this method only gives an inference speed of 4.7 fps, which is far below the desired real-time speed of 20 fps on standard hardware. We experimented with ResNet 50 on a bigger dataset like Tiny ImageNet [Appendix F] and the results are promising. It would be interesting to further extend it to ImageNet and to investigate whether these pruned backbone can be used with Siam-RCNN to achieve inference speed boost. We have added a similar discussion in section 6 of the paper.
>
> [4] Siam R-CNN: Visual Tracking by Re-Detection, CVPR, 2020
>
> ---
>
> **Q4.** There is a submitted paper (to this ICLR) that includes continuous relaxation of discrete network structure optimization for network growing (not pruning). “Growing Efficient Deep Networks by Structured Continuous Sparsification”. The direction of the paper is totally different, but there seems to be a similar part in terms of continuous relaxation. It would be good to mention the difference with this paper w.r.t. continuous relaxation.Also, I would like that more related works are mentioned in recent techniques using continuous relaxation(approximation).
>
> **Ans** Thank you for pointing us to this paper. It indeed seems an interesting direction. It uses hard sigmoid function as a relaxation of signum function. In contrast to this we use two continuous projection logistic and heaviside on top of one another which also helps in introducing crispness/sparsity in the masks. We have also experimented with such functions in early phases instead of our logistic function but we noticed that gradients become zero at the extremities, making it harder to push the values of $\psi$. Thank you for the suggestion. We will add papers related to continuous relaxation and others as suggested by other reviewers together.
>
> ---

---

### Official Review · AnonReviewer4 · 2020-10-28
**ChipNet: Budget-Aware Pruning with Heaviside Continuous Approximations**

**Rating:** 7
**Confidence:** 4

**Review:**

This paper proposes a new deterministic pruning strategy that employs continuous Heaviside function and crispness loss to identify a sparse network out of an existing dense network. Experiments show its effectiveness and robustness. Generally, it is well-written and easy to follow. Some minor issues are shown below.

1.	Pseudo codes for the functions mentioned in Algorithm 1 should be provided to show them clearly.
2.	The proposed algorithm is a deterministic algorithm and may fail in complex network pruning problems. An in-depth analysis of its limitation is needed.
3.	It is recommended to use different labels for different algorithms in the figures.

---

> ### Author Response · Authors · 2020-11-13
> **Response to Reviewer 4**
>
> Thank you for your positive feedback. We are very happy that you found our method effective and robust.
>
> **Q1:** Pseudo codes for the functions mentioned in Algorithm 1 should be provided to show them clearly.
>
> **A1:**  We have now updated the draft to include pseudo-codes or additional descriptions for each of the functions mentioned in Algorithm 1. These can be found in Appendix E in the updated draft.
>
>
> **Q2:** The proposed algorithm is a deterministic algorithm and may fail in complex network pruning problems. An in-depth analysis of its limitations is needed.
>
> **A2:** It could be possible that due to the deterministic nature of ChipNet, there may be few limitations associated with it and rigorous testing might be needed for full-proof reliability. In this regard, we already attempted to study ChipNet from different aspects. For example, we analyzed it with respect to different choices of budget constraints, performed stability tests for even extreme scenarios of as low as 1% parameters, analyzed how the masks get distributed across the network, and even studied the transferability of masks. For all these experiments, ChipNet has proved to work well. Some other experiments which are now running:
> - To check whether ChipNet is applicable for originally lightweight networks such as MobileNet, we are now running experiments on it as well. Based on how the results end up, we will add it to either the limitation or the benefit of the method.
> - Further, as suggested by other reviewers we are testing our method on a common and preferred choice of model, ResNet50 with a more complex dataset, TinyImageNet which will serve as a good approximation of ChipNet performance on ImageNet.
>
> These are only the additional directions that we could think of in terms of exploring the additional potential or limitation of ChipNet. We are adding the important points in a new Section titled ‘Limitations and Future Work’ of the paper. We will improve this section based on discussions with all the reviewers during the rebuttal phase. In case you have any other suggestions on limitation/possible future work, we would be happy to hear and if possible/feasible implement it in the paper.
>
> **Q3:** It is recommended to use different labels for different algorithms in the figures.
>
> **A3:** Thank you for the remark. We have now updated the labels for improved clarity.

---

> > ### Author Response · Authors · 2020-11-23
> > **Update to A2**
> >
> > All experiments have been completed and the results are added in Appendix F. We conclude that Chipnet is showing promising results for MobileNetv2 and on complex datasets like TinyImageNet. A summary of all the major changes is also provided as a separate comment in this forum.

---

### Official Review · AnonReviewer2 · 2020-10-30
**Incremental change to pruning methods with limited validation**

**Rating:** 6
**Confidence:** 3

**Review:**

This submission proposes a way to prune neural networks using a continuous penalty function.

== Pros ==

The experiments include some interesting illustrations beyond the basic accuracy comparisons. This includes illustrations of how weights are distributed between the layers (Fig 5), ablations on different components of the penalty function, and experiments on the transferability of the selected sparsity pattern.

There is a very complete description of the training procedure. Writing overall is clear. Code is included with submission. The code is readable and easy to get running.

== Cons ==

Many previous works propose a new sparsity penalty function, and associated optimization, for neural network training. As well as (Lemaire et al) and (Louizos et al) cited in the submission, there's is works such as:

  * Srinivas et al. "Training Sparse Neural Networks." CVPR 2017.
  * Manessi et al. "Automated Pruning for Deep Neural Network Compression." ICPR 2018.
  * Zhu et. al. "Improving Deep Neural Network Sparsity through Decorrelation Regularization." IJCAI 2018.
  * Yang et. al. "DeepHoyer: Learning sparser neural network with differentiable scale-invariant sparsity measures." ICLR 2020

Within this general class of approaches, though, the specifics of the penalty function and optimization in this paper are novel. Whether that is a valuable novelty depends entirely on the empirical results: since the higher-level idea is well-explored this adds to the literature iff the submission has made the best design choices among these other methods.

Despite their overall heft, the experiments have some missing pieces that make it hard to evaluate this. To start, the evaluation is on a less common (within the sparse neural network literature) choice of backbones for each dataset. See:

Blalock et. al. "What is the State of Neural Network Pruning?" Sections 3.3 & 4.2

Wider-ResNet would be expected to have greater redundancy than ordinary ResNet, perhaps making an easier problem for sparsifying. However, this choice is inherited from BAR.

Ordinary ResNet-50 is a more common choice, but less so on CIFAR-10. This choice differs from the corresponding experiments in the (Liu et al) work that is compared against, that uses VGGNet, DenseNet-40, and ResNet-164.

The code appears to have partial support for a much wider variety of models (see line 19 of pruning.py), though maybe not all are actually supported: for instance r32 and r152 don't seem to be included in the branches in models/resnet.py lines 421-428.

And generally there are few comparisons made. This is possibly unavoidable due to the overall fragmentation of metrics. Choices of comparisons are the more closely similar papers in the literature (this is good), but lacks more some of the simple baselines (that can often do surprisingly well):

Gale et. al. "The State of Sparsity in Deep Neural Networks"

I don't see experiments that show proposed optimization has consistent convergence across runs. Experiments dealing with "stability" are either highlighted single comparisons, or look at stability w.r.t. hyperparameters. Some of the reported accuracy differences in comparison are quite small, especially at higher levels of sparsity, so seeing the standard deviation of multiple runs of each method compared might be especially important.

== After Rebuttal ==

I had previously missed that Lemaire et. al. had some of the key comparisons (simple baselines done for the same architecture/dataset pairs) that are necessary to judge this method. I thank the authors for pointing this out. It is also helpful that these baselines are now included for completeness.

C.6 still seems like an indirect measure of optimization stability, but it is reasonable to conclude from it that the differences from BAR are real, given the very low standard deviation in accuracy.

I remain on the side that this is still a relatively incremental addition to the pruning literature, but I am now satisfied that it is well-validated.

---

> ### Author Response · Authors · 2020-11-19
> **Response to Reviewer #2 (1/2)**
>
> Thank you for your valuable comments. We address your questions and comments below.
>
> **Q1.** Many previous works propose a new sparsity penalty function, and associated optimization, for neural network training. As well as (Lemaire et al) and (Louizos et al) cited in the submission, there's is works such as:
>
> Srinivas et al. "Training Sparse Neural Networks." CVPR 2017. Manessi et al. "Automated Pruning for Deep Neural Network Compression." ICPR 2018. Zhu et. al. "Improving Deep Neural Network Sparsity through Decorrelation Regularization." IJCAI 2018. Yang et. al. "DeepHoyer: Learning sparser neural network with differentiable scale-invariant sparsity measures." ICLR 2020
>
> Within this general class of approaches, though, the specifics of the penalty function and optimization in this paper are novel. Whether that is a valuable novelty depends entirely on the empirical results: since the higher-level idea is well-explored this adds to the literature if the submission has made the best design choices among these other methods.
>
> **Ans** We agree with the reviewer that there exist several previous works that have proposed new sparsity penalty functions. We differentiate ourselves with these works on the following grounds motivating the reasons for each of these.
> 1. One of the primary novelties of this work that differentiates it from most other works is that ChipNet is easily integrable with multiple budget constraints without much hyper-parameter changes.
> 2. Another novel contribution to our design is the choice of two continuous functions for optimization - logistic and Heaviside. This allows us to devise the novel crispness loss term that plays a crucial role in guiding the optimization process towards 0-1 designs. Further, the multiobjective loss function and training in a continuous relaxation sense help our model to generally make more optimized choices on the number of parameters for any given choice of volume or channel constraint.
> 3. Unlike the existing works such as Network Slimming [1] and BAR [2], our design choice is free from hard-coding certain masks to cope up with instability issues for extreme pruning scenarios or other hard cases.
> 4. Further most existing methods belong to the category of unstructured pruning, while we focus on structured pruning in this paper. For example, among the example works outlined by the reviewer, the works of Srinivas et al 2017 [3] and Manessi et al 2018 [4] focus on unstructured pruning, as well as cannot be used with budget constraints. Further, Zhu et al 2018 [5] also do not impose budget constraints on the optimization. Their work studies the problem of sparsity in a direction orthogonal to ours. They propose decorrelation term in their loss function thereby inducing sparsity. We believe that our method of continuous relaxation combined with the decorrelation filters of Zhu et al [5] could provide a better sparsity regularization, however, this would be something to explore in future work. Our work is closest to Lemaire et al [2] in the sense of adding budget constraints. However, we show the sparsity behaviour with different choices of constraints and also demonstrate that with budget constraints, our method is superior. Further, our design complements well the budget constraints for various choices of model-data combinations.
> 5. Our continuous relaxation technique guides the optimization process towards a good locally optimal solution. This is demonstrated by our mask transfer results. We cannot argue whether it performs better than others since it was never explored before. However, we pave the groundwork to demonstrate that masks trained on richer datasets could be used for pruning during transfer learning.
> 6. Lastly, we do not need additional regularization as provided through knowledge distillation in Lemaire et al [2], and outperforms the latter even without using knowledge distillation.
>
>
>
> [1] Learning Efficient Convolutional Networks through Network Slimming, ICCV, 2014
>
> [2] Structured Pruning of Neural Networks with Budget-Aware Regularization, CVPR, 2019
>
> [3] Training Sparse Neural Networks, CVPR, 2017
>
> [4] Automated Pruning for Deep Neural Network Compression, ICPR, 2018
>
> [5] Improving Deep Neural Network Sparsity through Decorrelation Regularization, IJCAI, 2018

---

> > ### Author Response · Authors · 2020-11-19
> > **Reponse to Reviewer #2 (2/2)**
> >
> >
> > **Q2.**  Despite their overall heft, the experiments have some missing pieces that make it hard to evaluate this. To start, the evaluation is on a less common (within the sparse neural network literature) choice of backbones for each dataset. See:
> >
> > Blalock et. al. "What is the State of Neural Network Pruning?" Sections 3.3 & 4.2
> >
> > Wider-ResNet would be expected to have greater redundancy than ordinary ResNet, perhaps making an easier problem for sparsifying. However, this choice is inherited from BAR.
> >
> > Ordinary ResNet-50 is a more common choice, but less so on CIFAR-10. This choice differs from the corresponding experiments in the (Liu et al) work that is compared against, that uses VGGNet, DenseNet-40, and ResNet-164.
> >
> > **Ans** We agree with the reviewer that to demonstrate the full proof efficacy of a pruning method, experiments on a wider set, and preferably harder problems, are desired. We tried to achieve this through the experiments shown in the paper. Our choice of WideResNet was motivated from the state-of-the-work of BAR. Further, based on recommendation and feasibility, we now also add experiments on ResNet50 with TinyImageNet and experiment on MobileNetv2 as well. We further add experiments on pruning ResNet110 for C10, based on a point raised by another reviewer to make a comparison with DeepHoyer [1]. Related results are presented in Section F of the updated draft.
> >
> > [1] DeepHoyer: Learning Sparser Neural Network with Differentiable Scale-Invariant Sparsity Measures, ICLR 2020.
> >
> > ---
> > **Q3.** The code appears to have partial support for a much wider variety of models (see line 19 of pruning.py), though maybe not all are actually supported: for instance r32 and r152 don't seem to be included in the branches in models/resnet.py lines 421-428.
> >
> > **Ans** Thank you for the remark. We have now added several other variants Resnet 18, 20, 32, 50, 56, 101, 152 in the code. Further, we added MobileNetv2 as well.
> >
> > ---
> >
> > **Q4.** And generally there are few comparisons made. This is possibly unavoidable due to the overall fragmentation of metrics. Choices of comparisons are the more closely similar papers in the literature (this is good), but lacks more some of the simple baselines (that can often do surprisingly well):
> >
> > Gale et. al. "The State of Sparsity in Deep Neural Networks"
> >
> > **Ans** Indeed as pointed out by the reviewer, our comparisons are mostly directed towards the works that are closely related. We specifically focused on methods that could provide a control on the use of resources, therefore, allowing to directly or indirectly include a resource constraint in the optimization process. Further, since BAR was already shown to work on several simpler baselines we removed the scores for the latter. However, for the sake of completeness, we now also add a table of results in Appendix C.2 where comparisons with simpler baselines including ID [1], WM [2], and Random baseline are now presented. ChipNet outperforms all these baselines.
> >
> > [1] Exploiting linear structure within convolutional networks for efficient evaluation.NIPS, 2014.
> >
> > [2] Learning both weights and connections for an efficient neural network. NIPS, 2015
> >
> > ---
> >
> > **Q5.** I don't see experiments that show proposed optimization has consistent convergence across runs. Experiments dealing with "stability" are either highlighted single comparisons, or look at stability w.r.t. hyperparameters. Some of the reported accuracy differences in comparison are quite small, especially at higher levels of sparsity, so seeing the standard deviation of multiple runs of each method compared might be especially important.
> >
> > **Ans**  ChipNet is a purely deterministic method and there is no stochastic component in the pruning or finetuning step. Also, we would like to point out that we use the same value of all the four hyperparameters throughout all the experiments reported in the paper. For the pretraining step of the dense network, we use standard procedure, and the final results of pretraining do not generally vary across runs.
> >
> > We are also extending our study to show how sensitive the method is towards perturbation in these hyperparameters. We performed 5 runs of pruning of WRN model on CIFAR 100 dataset at 16x volume pruning factor where the 4 hyperparameters $\alpha_1$, $\alpha_2$, $\beta_{inc}$, $\gamma_{inc}$ where chosen within +-10% perturbations of the base values, and the results are now presented in Appendix C.6 of the paper. For this case, we found the deviation is comparable to that of BAR with around 1% improvement in mean accuracy over the latter. We believe that this experiment is very relevant to the main text. Based on this, we will run more cases in the future and update the draft accordingly.
> >
> > ---

---

### Author Response · Authors · 2020-11-19
**Summary of Major Changes**


Thanks to all the reviewers for their constructive reviews and feedback. Here is a summary of the changes



**Writing**

-   Added Section 6 discussing the limitations and future work for ChipNet

-   Added Appendix C.6 to show sensitivity analysis

-   Added Appendix E providing the pseudo code for functions used.

-   Added Appendix F for additional experiments related to Resnet 50 Tiny ImageNet, MobileNetV2 and comparison with DeepHoyer.

-   Added more baselines for comparison in Table 5.




**Code**

-   Added more variants of ResNet.

-   Added support for MobileNetV2

---

### Decision · Program_Chairs · 2021-01-07
**Final Decision**

**Decision:**

Accept (Poster)

**Comment:**

 This paper proposed a new method to prune neural networks using a continuous penalty function. All reviewers suggest acceptance (some are on borderline though) as the authors did a good job in the rebuttal phase. AC also could not find any particular reason to reject the paper (in particular, the overall writing is clear) and thinks that this paper is a meaningful addition to ICLR 2021.